# Exploring shifts in values among urban Senegalese: The impact of global crises on social and cultural norms

Yachiyo Tobita○[1]*, Mandiaye Diagne[2], Joseph Bassama[3], Moussa Ndong[3], Mor Gueye[4], Kiyokazu Ujiie[5]

1 Graduate School of Science and Technology, Degree program in Life and Earth Sciences, University of Tsukuba, Tsukuba, Ibaraki, Japan, 2 Africa Rice Center, Sahel Regional Station, Saint-Louis, Senegal, 3 Laboratory LABAAM, Gaston Berger University, Saint-Louis, Senegal, 4 Department of Science Agronomy, Fisheries and Food Technology, Gaston Berger University, Saint-Louis, Senegal, 5 Faculty of Life and Environmental Sciences, University of Tsukuba, Tsukuba, Ibaraki, Japan

* s2030239@u.tsukuba.ac.jp

**Data Availability Statement:** The relevant data and R script for analysis and visualization are available in the Support Information. The open-source

## Abstract

The COVID-19 pandemic triggered social and economic stagnation worldwide, significantly impacting people's lives. In addition, the Russia-Ukraine war that began in 2022 resulted in rising food prices globally, severely affecting low- and middle-income countries. This study aimed to examine the impact of these unprecedented crises on individual values, focusing on Senegal's urban population. This study is the first to quantitatively assess changes in the values of urban Senegalese during this global crisis. Surveys were conducted in Saint-Louis, Senegal, in August-September 2018 and June-July 2022. The timing of these studies coincides with the onset of the COVID-19 pandemic in early 2020 and the outbreak of the Russia-Ukraine war in February 2022. The findings revealed a 19.9% decrease in the average monthly cost of living per capita between 2018 and 2022, attributed to the combined effects of rising food prices and unemployment. Furthermore, the proportion of households spending less than $3.50 per person per day—below the lower-middle-income class poverty line—increased by 11.05%. Our analysis indicates a decline in values such as benevolence, universalism, hedonism, and self-direction. In contrast, values related to power and achievement significantly increased following the pandemic. These results suggest that individual values are flexible and may change in response to external factors such as global crises.

## Introduction

The coronavirus disease (COVID-19) resulted in wide-ranging and severe impacts across the globe. The global economy slumped, leading to an economic crisis often referred to as the "Great Lockdown." This crisis has had particularly severe social and economic repercussions in Africa, where the social infrastructure is relatively weak. Approximately 30 million people in Africa have fallen into extreme poverty due to the pandemic [1], and 278 million individuals in the region are now food insecure in the wake of economic stagnation [2]. Senegal was

datasets used in Table 4 is available at: WVS Database (worldvaluessurvey.org).

**Funding:** The study in 2018 was supported by public interest incorporated foundation, Urakami Foundation for Food and Food Culture Promotion (https://www.urakamizaidan.or.jp/en.html), grant number: 2017-30 (K.U.) and public interest incorporated foundation, Food Culture Promotion and Ajinomoto Foundation for Dietary Culture (https://www.syokubunka.or.jp/english/), grant number: 2017-01(K.U.). The study in 2022 was supported by JST-SPRING, Japan Science and Technology Agency (https://www.jst.go.jp/EN/), grant number: JPMJSP2124(Y.T.). The funders had no role in study design, data collection and analysis, decision to publish, or preparation of the manuscript.

**Competing interests:** The authors have declared that no competing interests exist.

among the first francophone West African countries to be affected by COVID-19, implementing early infection control measures [3].

The consequences of social immobility are manifold. The Senegalese government reported that movement restrictions associated with COVID-19 increased social unrest, leading to a significant rise in theft and land disputes among farmers [4], as well as an increase in the unemployment rate [5]. Additionally, import rice prices have risen, as shown in Fig 1 [5]. As the world began to recover from this economic downturn, the Russia-Ukraine war erupted in 2022, sharply increasing food and fuel costs and raising concerns about the long-term economic impacts on low- and middle-income countries [6].

The pandemic is likely to have lasting effects on the mental health and well-being of those impacted by social restrictions, economic consequences, and fear of infection [7–9]. In Africa, the pandemic has resulted in income losses and increased psychological distress due to concerns about inadequate access to food and basic necessities [10]. In addition, regional and cultural differences in countries significantly influence people's attitudes and behaviors toward infection globally [11]. A comparative study of West African countries found that many individuals expressed greater concern about the health risks associated with COVID-19 vaccination than the actual risk of infection, which has complicated the implementation of preventive measures [12].

Research has utilized quantitative value indicators to examine how external factors, such as global economic crises, influence changes in human values. Significant events such as the global financial crisis, wars, and terrorism have notably impacted human values both economically and socially [13–15]. Studies on changes in human values during the current pandemic have been conducted in several countries [7, 16, 17], demonstrating that external factors can influence human motivation and behavior. Previous research indicates that during crises such as wars and pandemics, individuals tend to prioritize conservative values driven by heightened fear and restrictive measures [7, 13, 16, 17].

Understanding how people's values change and which behavioral tendencies are reinforced during significant environmental changes is imperative, especially when implementing behavioral and other restrictions in future crises. However, research on the psychological impacts of global events on value changes is limited in Africa, where economic vulnerability necessitates careful crisis management. Changes in human values may also vary depending on the socioeconomic context [14], highlighting the need to measure differences in value changes based on cultural and regional factors.

This study aims to assess changes in values among urban residents by analyzing surveys conducted in 2018 and 2022. Specifically, we seek to understand how the socioeconomic shifts caused by the COVID-19 pandemic and the Russia-Ukraine war influenced the values of the urban Senegalese population. During this period, several global events unfolded, including the emergence of COVID-19, the outbreak of the Russia-Ukraine war, and the resulting global economic turbulence. By comparing values across these two time periods, we aim to shed light on how these historic global events impacted urban Senegalese values.

There is a notable lack of quantitative studies focusing on the detailed psychological changes among urban Senegalese populations. The novelty of this study lies in its quantitative assessment of changes in values during this period for a specific urban population in Africa, using a standardized value indicator that allows for comparison with other countries and time periods.

The structure of this paper is as follows: First, we provide an overview of Schwartz's theory of values, which serves as the study's theoretical framework. Subsequently, we review studies on individual changes due to COVID-19, using value indicators to highlight the novelty of our research. In the methodology section, we detail the original data collection process. While

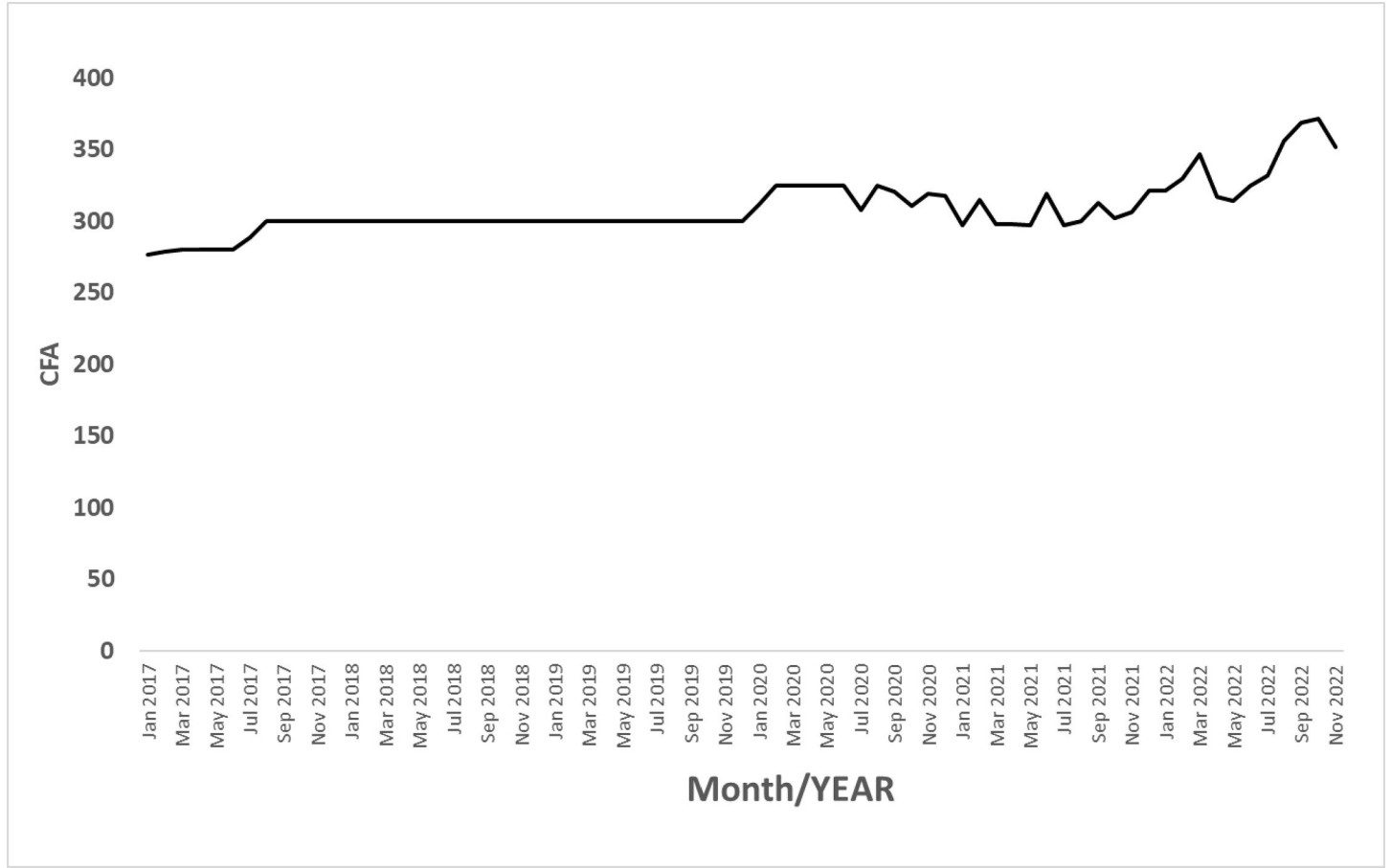

**Fig 1. Change in import rice prices in Saint-Louis(2017–2022).** Note: This figure is based on the rice prices in Saint-Louis downloaded from ANSD(https://www.ansd.sn/).

previous studies indicate that the pandemic led to an increased emphasis on conservative values, the shift in value orientation among urban Senegalese people demonstrated a somewhat different trend. The discussion section analyzes the psychological changes observed in light of our survey findings. Finally, in the conclusion section, we discuss the factors that influenced value changes in Senegalese people, address the limitations of this study, and suggest areas for future research. We believe that by assessing the changes in individual values observed in urban Senegal as a result of the global crises, this study will contribute meaningfully to the growing body of research in this area.

## Human values

Values determine what people consider important in life and shape their broad motivational goals [18]. Therefore, values influence everyday decision-making and actions [19]. Schwartz's theory of basic values has become a widely adopted framework in value-related research [20]. His value items are commonly used as indicators to assess individual values and are featured in the World Values Survey (WVS) [21], a representative example of international value research.

Research on basic values has sought to identify how values are shaped not only by factors such as age, gender, and occupation but also by contextual influences such as communities,

political systems, and religion [18–20]. This body of work examines human behavioral motivations, with Schwartz's values applied across various research areas, including food consumption analyses, cross-cultural comparisons, environmental protection priorities, and the psychological impact of social change [22–26]. For example, individuals adapt to new circumstances by adjusting their values in response to anxiety caused by environmental changes [26]. Comparative studies have also shown common patterns in value orientations across different cultures [24, 25], revealing a complex interplay between cultural norms, customs, and socioeconomic conditions that shape individual values.

### Content of values

As shown in Fig 2, Schwartz's theory of basic values outlines ten core human value orientations, which are grouped into four higher-order dimensions [20]. Conservation (tradition, security, conformity) emphasizes social order, stability, and the preservation of traditional customs. Openness to change (self-direction, stimulation, hedonism) reflects adaptability to

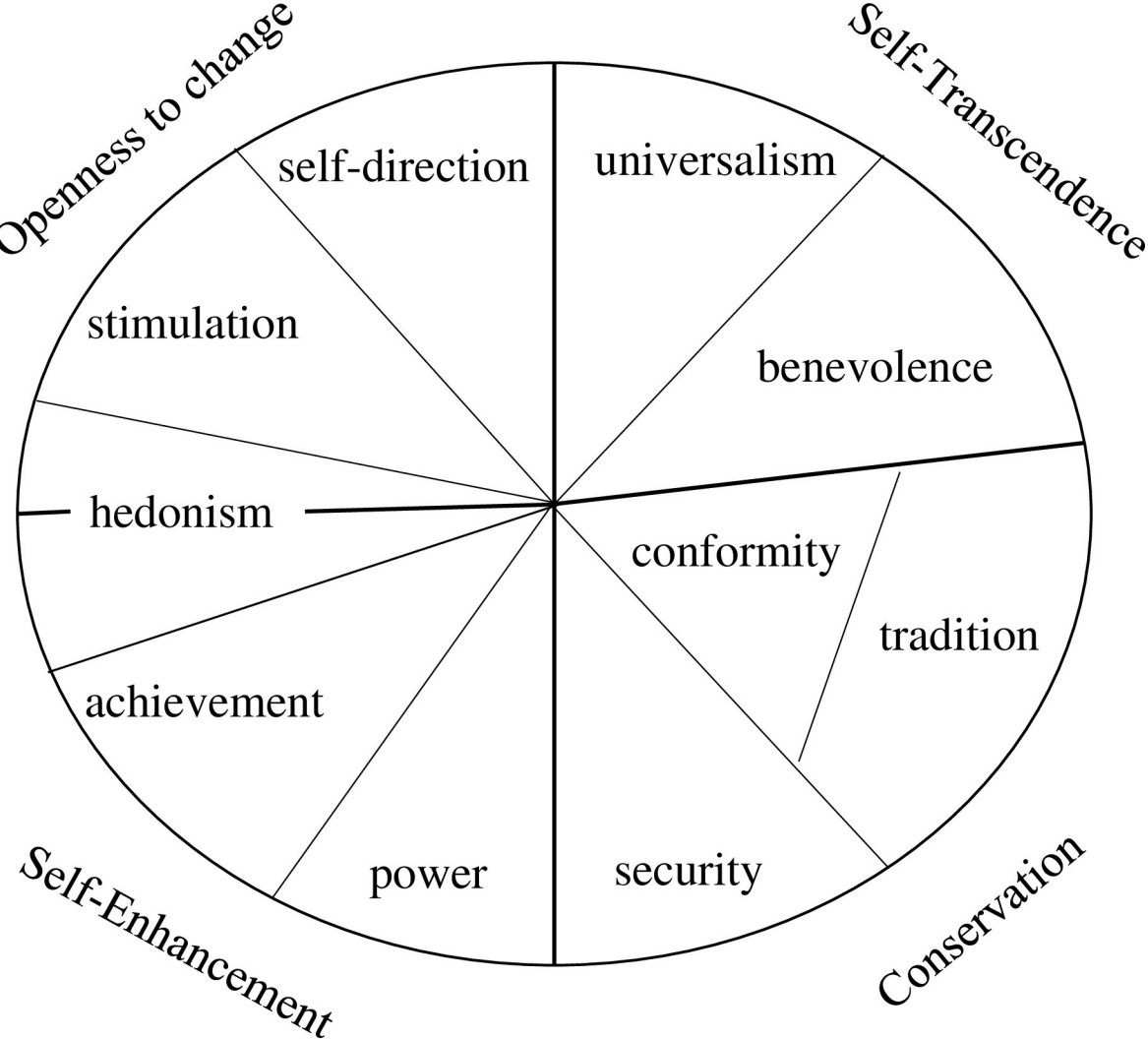

**Fig 2. Theoretical model of the structure of relationships among ten motivational types of value.** Note: This figure based on Schwartz [20] that licensed under Creative Commons Attribution-Noncommercial-No Derivative Works 3.0 License.

changing circumstances and opportunities, characterized by a focus on independence, creativity, self-reliance, and a willingness to take risks. It stands in contrast to the values of conservation. Self-transcendence (universalism, benevolence) involves a focus on the well-being of others, including promoting environmental sustainability and social welfare. Self-enhancement (power, achievement) prioritizes self-interest, financial success, and social status, with an emphasis on personal gain over the interests of others.

In the theoretical model, hedonism is positioned between openness to change and self-enhancement; however, in this study, it is placed within the dimension of openness to change.

## Impact of social change on people's values

While values tend to be relatively stable, they can shift in response to significant life events and external circumstances [22]. Numerous studies have examined how rapid changes in the external environment influence individual values. It has been observed that when people perceive risks or threats, values related to self-conservation (e.g., security and tradition) become more important, whereas the significance of values related to self-initiated change and opportunity (e.g., stimulation and self-direction) tends to decrease. These patterns of value shifts have been documented in studies of wars, terrorism, and global financial crises [13–15].

For example, a study of young Israelis used a 44-item version of the Schwartz scale to examine short-term value changes during and after the 2006 Israel-Lebanon war. The results showed that during the war, the importance of values such as tradition, power, and security increased, while the importance of benevolence, universalism, self-direction, stimulation, and hedonism decreased [13]. In contrast, Verkasalo et al. [15] used a ten-value framework to examine changes in the values of Finnish adolescents after the 9/11 terrorist attacks. The importance of security increased significantly following the attack but returned to baseline shortly thereafter. This study compared these shifts across young adults and university students, as well as male and female participants, and found no significant differences in value changes between the groups.

Using data from the European Social Survey (2002–2014), Sortheix et al. [14] analyzed changes in the values of young people in 16 European countries before and after the 2008 financial crisis. During the crisis, the importance of security, tradition, and benevolence increased, whereas the importance of hedonism, self-direction, and stimulation declined. However, changes in value priorities among young people varied based on youth unemployment rates and social security spending. In countries with higher social security investment, values shifted differently than in countries with lower investment. These disparities were largely attributed to differences in the availability and implementation of support systems, such as unemployment benefits. In countries with high social security investment, values emphasizing social order and stability—such as security, tradition, and benevolence—became more important during the crisis. In contrast, in countries with low social security investment, values linked to self-interest, financial success, and social status—such as power, achievement, and conformity—were considered critical. These studies illustrate that major events such as war, terrorism, and financial crises can rapidly alter individual values, often leading to an increased emphasis on security.

The COVID-19 pandemic is another significant event that has influenced human values. Since the onset of COVID-19, online surveys conducted among adults in various countries have documented value changes in response to the infection and control measures such as movement restrictions and social distancing. In Poland, for example, Schwartz's 19 value items were used to identify the impact of the pandemic on well-being. The results indicated an increased importance of self-direction, security, conformity, and universalism, while the

significance of hedonism declined. This suggests that people adapt to new circumstances by adjusting their values in response to external social changes [7].

A study in France investigated how the threat of infection influenced attitudes toward compliance with new lifestyle changes, such as behavioral restrictions and social distancing, during the pandemic. The results showed that conservation values (e.g., conformity and security) increased, whereas stimulation values, which favor change, decreased. This shift was significantly associated with behaviors that aligned with government restrictions and policies encouraging social distancing, underscoring the importance of health and economic stability as key factors in value changes [16].

A longitudinal study of preferred values in Australia utilizing Best Worst Scaling (BWS) expanded the methodology developed by Lee et al. [27, 28]. This approach was further explored by Daniel et al. [17] and Sneddon et al. [29]. Daniel et al. [17] analyzed 19 value items using BWS and found that the pandemic heightened anxiety among the population, leading to increased conservative tendencies. This shift led to a significant retreat from values that prioritize social acceptability, creativity, self-expression, compassion for others, and the conservation of nature. Sneddon et al. [29] conducted pre- and post-pandemic surveys on individual values related to sustainable environmental activities using the BWS methodology. They found that the importance attached to values promoting environmental sustainability increased before the pandemic but decreased afterward. Notably, individuals with daily experiences in nature exhibited only a marginal decline in the importance of sustainable environmental practices, even during the pandemic. Both studies indicated that changes in values differed significantly from the beginning to the end of the pandemic. However, most of these studies have been conducted in wealthy countries. Given the significant differences in socioeconomic conditions, particularly among high-, low-, and middle-income countries, further research encompassing diverse regions is imperative for a comprehensive understanding of this subject.

## Methods

### Values questionnaire

In this study, we assessed participants' values using survey questions from the WVS Wave 6 (2014–2017), based on Schwartz's ten dimensions value dimensions [21]. This version included two questions derived from the benevolence dimension to evaluate individuals who prioritize "helping or making those around them happy" or "contributing to society." Respondents were asked either one or both of these questions [21, 30]. Given the social context of Senegal, where many people live in extended families, this study utilized these two options to measure whether respondents preferred to value kindness toward their close family members or the broader community. Schwartz et al. [31] referred to the values reflecting kindness toward close individuals as "benevolence-caring" and those reflecting kindness toward society as "benevolence-dependability." In this study, we follow Schwartz's notation to distinguish between these two values. Therefore, this study adopts 11 value items, as listed in Table 1.

The value indicators were developed as a standardized method for quantitatively measuring the psychological scales of individuals with varying cultural backgrounds. Numerous international comparative studies, including the WVS, have employed these indicators. The primary advantage of employing these indicators is their standardization, which facilitates future international or intertemporal comparisons. Notably, there is a dearth of value surveys focused on West African countries, and this study aims to contribute to advancing research in this area.

**Table 1. Questions for each value attribution based on Schwartz's theory of basic values.**

| Value Attribution[a] | Question[b] |
|---|---|
| *Conservation* | |
| Tradition: Respect, commitment and acceptance of the customs and ideas that traditional culture or religion provide the self | Tradition is important to this person; to follow the customs handed down by one's religion or family |
| Conformity: Restraint of actions, inclinations, and impulses likely to upset or harm others and violate social expectations or norms | It is important to this person to always behave properly; to avoid doing anything people would say is wrong |
| Security: Safety, harmony and stability of society, of relationships, and of self | Living in secure surroundings is important to this person; to avoid anything that might be dangerous |
| *Openness to change* | |
| Self-direction: Independent thought and action-choosing, creating, exploring | It is important to this person to think up new ideas and be creative; to do things one's own way |
| Hedonism: Pleasure and sensuous gratification for oneself | It is important to this person to have a good time; to "spoil" oneself |
| Stimulation: Excitement, novelty, and challenge in life | Adventure and taking risks are important to this person; to have an exciting life |
| *Self-Transcendence* | |
| Universalism: Understanding, appreciation, tolerance and protection for the welfare of all people and for nature | Looking after the environment is important to this person; to care for nature and save life resources |
| [c]Benevolence-dependability: Preservation and enhancement of the welfare of society | It is important to this person to do something for the good of society |
| [c]Benevolence-caring: Preservation and enhancement of the welfare of people nearby | It is important for this people to help the people nearby; to care for their well-being |
| *Self-Enhancement* | |
| Power: Social status and prestige, control or dominance over people and resources | It is important to this person to be rich; to have a lot of money and expensive things |
| Achievement: Personal success through demonstrating competence according to social standards | Being very successful is important to this person, to have people recognize one's achievements |

Note:[a]Definition of Schwartz'value of theory is based on Schwartz[20].

[b]Question is based on World Value Survey[21]

[c]Definition of Benevolence-dependability and Benevolence-caring are based on Schwartz and Ciciuch[31]

## Measures

We used BWS, specifically Case 1, to evaluate respondents' values. BWS is a survey method designed to measure people's preferences or the relative importance they assign to multiple items. It has been widely applied in various academic fields in recent years, including health, psychology, and agricultural economics [17, 28, 29,32–36].

In a BWS survey, a subset of all items is presented to respondents, who then choose the most important (best) item and the least important (worst) item from that set. In addition, respondents select one attribute within each question as either the best or worst, making it easier for them to understand and engage with the question [34, 36]. BWS has also been used in surveys based on Schwartz's theory of basic values [17, 27, 28]. For example, Lee et al. [27] quantitatively measured Schwartz's ten value items using BWS and highlighted that traditional self-report ratings often suffer from the problem of respondents applying different evaluation criteria. BWS helps reduce this effect by providing a more structured comparison. According to Lee et al. [27], BWS is an effective method for revealing people's values because it is both more straightforward and relative than traditional methods, such as the Likert scale. Furthermore, Lee et al. [28] conducted a BWS assessment of 19 value items, including new items such as animal welfare, in samples of adults from Australia and the US, demonstrating the utility

and reliability of this approach. Daniel et al. [17] used BWS to examine the impact of the pandemic on individual values, employing the counting approach to determine the relative importance of different factors. Similarly, Sneddon et al. [29] used the BWS counting approach to explore changes in individual values related to the natural environment during the pandemic. The results of these studies suggest that the BWS is an effective approach for capturing changes in individual values.

To design the survey, we used the support.BWS package [37] in R Core Team version 4.2.2. [38] to create BWS Case 1 questions. The question design followed a Balanced Incomplete Block Design (BIBD), implemented using the crossdes package [39]. Table 2 shows the combinations of value items, each randomly displayed six times using BIBD. As shown in Fig 3, respondents were presented with 11 questions, each consisting of six items, and were asked to select one attribute from the six that was either "very much like me" or "not at all like me."

## Preferences and data analysis

Preferences for the items were measured using a counting approach to calculate Best-Worst (BW) and standardized scores, as in previous studies. The counting approach represents the difference between the number of times respondents selected an item as best (B) or worst (W), as shown in Eq (1):

$$BW_{n,i} = B_{n.i} - B_{n.j} \tag{1}$$

where $BW_{n,i}$ represents the BW score for item $i$ for respondent $n$, $B_{n.i}$ represents the number of times respondent $n$ chose item $i$ as the most preferable, and $B_{n.j}$ represents the number of times $j$ was chosen as the least preferable.

Furthermore, the standardized BW score, expressed in Eq (2), adjusts the BW score for each item on a scale of -1 to +1. Here, $f_i$ represents the number of times an item was presented in the survey.

$$std.BW_{n,i} = \frac{\overline{BW_i}}{f_i} \tag{2}$$

The dataset preparation and analysis were conducted using R software [38] and the support.BWS package [37].

**Table 2. Combinations of displayed value items in the questionnaire.**

| Value question | Q1 | Q2 | Q3 | Q4 | Q5 | Q6 | Q7 | Q8 | Q9 | Q10 | Q11 |
|---|---|---|---|---|---|---|---|---|---|---|---|
| Tradition | 0 | 0 | 1 | 1 | 1 | 1 | 1 | 0 | 0 | 1 | 0 |
| Conformity | 0 | 1 | 0 | 1 | 1 | 1 | 0 | 0 | 1 | 0 | 1 |
| Security | 1 | 0 | 1 | 1 | 0 | 1 | 0 | 1 | 0 | 0 | 1 |
| Self-direction | 1 | 0 | 1 | 0 | 1 | 0 | 1 | 0 | 1 | 0 | 1 |
| Hedonism | 0 | 0 | 0 | 1 | 0 | 0 | 1 | 1 | 1 | 1 | 1 |
| Stimulation | 0 | 1 | 1 | 0 | 1 | 0 | 0 | 1 | 0 | 1 | 1 |
| Universalism | 1 | 1 | 0 | 0 | 0 | 1 | 1 | 0 | 0 | 1 | 1 |
| Benevolence-dependability | 1 | 0 | 0 | 0 | 1 | 1 | 0 | 1 | 1 | 1 | 0 |
| Benevolence-caring | 1 | 1 | 1 | 1 | 0 | 0 | 0 | 0 | 1 | 1 | 0 |
| Power | 1 | 1 | 0 | 1 | 1 | 0 | 1 | 1 | 0 | 0 | 0 |
| Achievement | 0 | 1 | 1 | 0 | 0 | 1 | 1 | 1 | 1 | 0 | 0 |

Note:1 indicates that each value item appears in the question.This combination of value items applies to the survey.

Now, I will briefly describe some people. Would you please indicate for each description whether that person is **Very much like you**, or **Not at all like you**?

| Very much like me | | Not at all like me |
|---|---|---|
| ☐ | It is important to this person to think up new ideas and be creative; to do things one's own way. | ☐ |
| ☐ | Living in secure surroundings is important to this person; to avoid anything that might be dangerous. | ☐ |
| ☐ | It is important for this people to help the people nearby; to care for their well-being. | ☐ |
| ☐ | Being very successful is important to this person, to have people recognize one's achievements. | ☐ |
| ☐ | Adventure and taking risks are important to this person; to have an exciting life. | ☐ |
| ☐ | Tradition is important to this person; to follow the customs handed down by one's religion or family. | ☐ |

**Fig 3. Example of a choice set for Best Worst Scaling.**

## Survey design

This study compares two surveys conducted in 2018 and 2022 among residents of Saint-Louis, Senegal, located 278 km from Dakar, the capital of Senegal. Africa's demographic transition and rapid urbanization are significantly impacting the lives of its people. Given our emphasis on evolving individual values in urban Africa, we focused on Senegal, which has the highest urbanization rate in West Africa. Saint-Louis is Senegal's fifth-largest state, with a population of approximately 1,120,585 and an area of 19,241 square kilometers [5]. According to the latest census, Saint-Louis has 27,517 households across 33 districts, with a population of 209,752 [40]. Fig 4 shows the survey location in Saint-Louis.

The first survey took place before the pandemic, from late August to early September 2018. After discussions with local collaborators, nine representative districts in Saint-Louis were selected for the study. A random walk method [41] was employed to select six to seven households per district, resulting in a sample of 60 households.

The second survey was conducted between the post-pandemic period and the Russia-Ukraine War, from early June to late July 2022, using the same random walk method. Of the 33 districts, 10 were selected using probability proportional to size sampling, based on census data provided by the Agence National de la Statistiqué et de la Démographie(ANSD) [40]. A total of 20 households were selected from each district, yielding a total sample of 200 households [42].

Students from Gaston Berger University served as investigators for both the 2018 and 2022 surveys, following the same procedures to conduct interviews. While the 2018 survey focused on households living on the sandbank islands of the Senegal River and in commercial and working-class districts, the 2022 survey was expanded to cover a broader urban population of

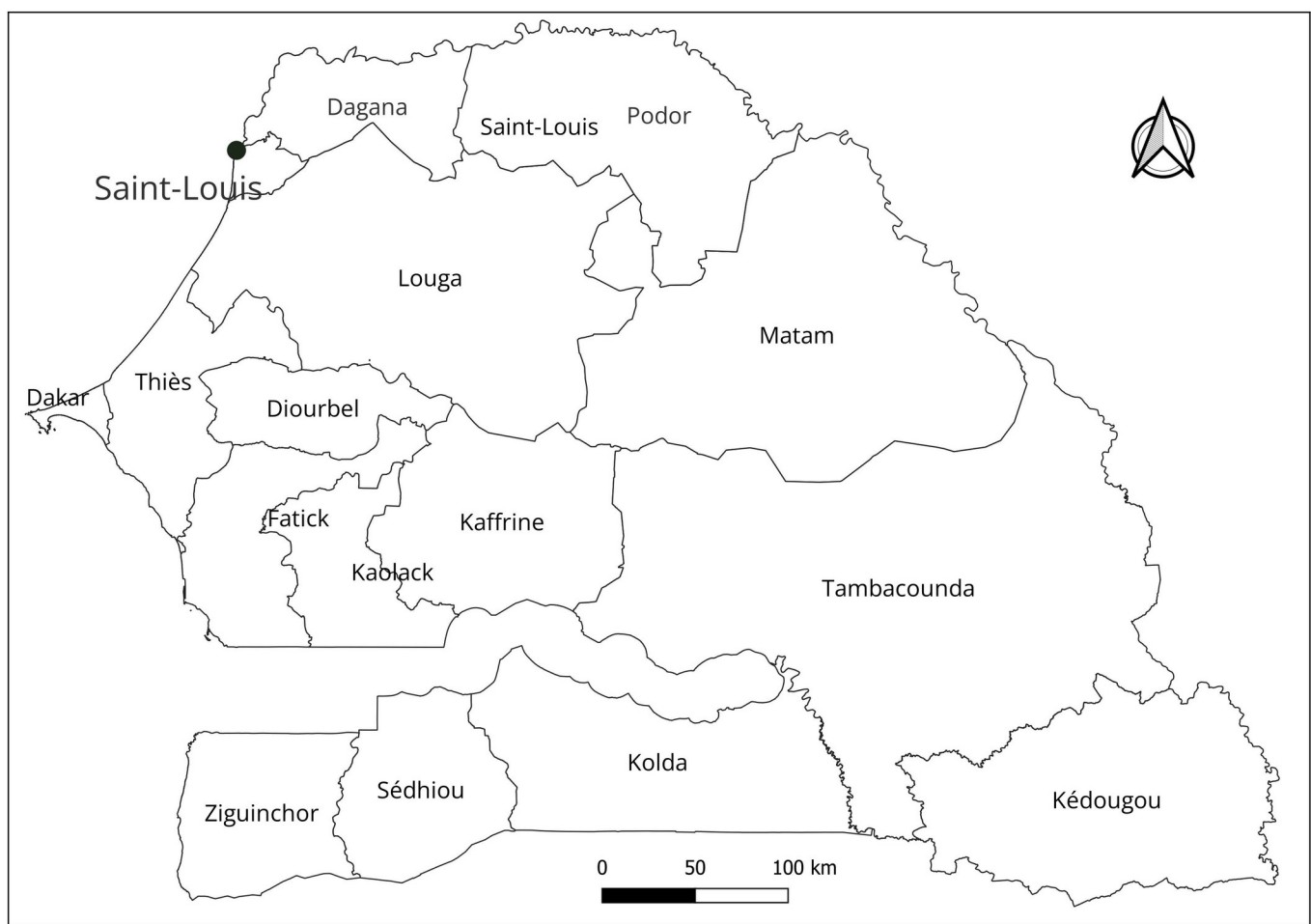

**Fig 4. Survey location in Senegal.** Note: The map is created with QGIS Development Team,2024. QGIS Geographic Information System. Open Source Geospatial Foundation. URL http://qgis.org.

Saint-Louis, including diverse neighborhoods such as fishing, commercial, and working-class districts.

In both surveys, the random walk method was used to select households, starting at intersections in the central district areas, with travel directions randomized. Only one household was selected per street, avoiding extended family members, relatives, neighbors, or friends of the participants. The person in charge of cooking or purchasing food in the household was selected as the survey respondent. The purpose of the study was explained to the respondents in advance, and they were assured that their personal information would remain anonymous and be used solely for scientific purposes. The survey collected demographic data and included a BWS-based value survey conducted in both French and Wolof. Since women primarily manage food buying and meal preparation in Africa [43], the majority of respondents were females aged 16 and above (Note 1). Written consent was obtained before the survey began.

The comprehensive research permissions, including ethical approval for both surveys, were obtained from relevant authorities. Ethical approval for the 2018 survey was granted by the Ministry of Higher Education, Research and Innovation in Senegal (Approval No: 0000303). Ethical approval for the 2022 survey was obtained from the Faculty of Life and Environmental Sciences at the University of Tsukuba (Approval No: 2022–2).

## Inclusivity in global research

Additional information regarding the ethical, cultural, and scientific considerations for inclusivity in global research can be found in the Supporting Information (SX Checklist).

## Characteristics of the survey sample

Table 3 provides a summary of the descriptive statistics of household characteristics for the 2018 and 2022 surveys, encompassing respondents' age, marital status, education, and employment. A comparison of 2018 (N = 60) and 2022 (N = 200) samples reveals that more

**Table 3. Descriptive statistics of respondents in the 2018 and 2022 surveys.**

| Variables | 2018 | 2022 |
|---|---|---|
| Total number of respondents | 60 | 200 |
| Age(Mean) | 42 | 43.6 |
| Marital status(Mean) | 76.67 | 75.00 |
| Religion(% of Muslim) | 100 | 100 |
| *Education* | | |
| Number of years in school | 7.1 | 5.33 |
| Never attended school(%) | 18.33 | 28.50 |
| Attended primary school and above (%) | 78.33 | 65.00 |
| Attended Koranic school(%) | 3.33 | 6.50 |
| *Occupation* | | |
| Housewife(%) | 55.00 | 51.00 |
| Small business(%)[a] | 30.00 | 35.00 |
| Government(%) | 6.67 | 3.50 |
| Office(%) | 3.33 | 5.00 |
| Agri-food(%) | 0.00 | 2.00 |
| Student(%) | 3.33 | 1.50 |
| Other(%) | 1.67 | 2.00 |
| Household size(Mean) | 8.90 | 11.29 |
| *Classified by amount of expenditure(per day per capita)* | | |
| Mean of daily consumption expenditure per capita(FCFA)[b] | 1,672 | 1,339 |
| US$ 2.15 | 3.33 | 3.95 |
| US$ 3.65 | 18.33 | 29.38 |
| US$ 6.85 | 58.33 | 76.84 |
| Rate of meal sharing (%)[c] | 23.16 | 33.90 |

Note:The first survey was conducted from August to September 2018 and the second survey was conducted from June to July 2022.

[a]Small business refer to selling food and goods or other self-employment such as craftman, tailor.Other is unemployed

[b]Daily consumption expenditure per capita is calculated by dividing total monthly household consumption expenditure by the number of household members and 30 days.Minor repair expenses is included(ANSD,2021). International poverty line is 519.8CFA or US$2.15 (2017 PPP) per day per capita; lower-middle-income class poverty line is 882.5 CFA or US$3.65 (2017 PPP) per day per capita;upper-middle-income class poverty line is 1656.2 CFA or US$6.85 (2017 PPP) per day per capita(World Bank 2023).The nationwide values for Senegal is 9.3% of the international poverty line, 34.7% of the lower-middle-income class poverty line and 74.4% of upper-middle-income class poverty line(World Bank 2018).

[c]The rate of sharing meals with friends,neighbors,and others non-household members is calculated by the frequency of lunch and dinner over a five days period.

respondents in 2022 (28.5%) reported having no formal education compared to 2018 (18.33%). The percentage of respondents who attended Koranic school was 3.33% in 2018 and 6.5% in 2022, indicating that these individuals were educated exclusively in private institutions run by Islamic leaders. No significant differences were observed between the two surveys in terms of religion, employment, or average age. Over 30% of respondents were employed in the informal sector, engaging in small businesses such as selling food from their homes.

The daily per capita consumption in 2018 was 1,672 FCFA, which dropped to 1,339 FCFA in 2022. According to the ANSD [44], the average daily consumption expenditure in urban areas, including Dakar, is 1,818 FCFA, while in rural areas, it is 1,040 FCFA, which is a reasonable value given that the survey was conducted in a rural city. The average household size was 8.9 members in 2018 and 11.3 in 2022, closely aligning with the 10.4 members reported by ANSD for urban households [45]. This consistency indicates that the sample is representative of the study population.

The daily consumption expenditure per person was also examined based on the World Bank's definitions of poverty lines. The World Bank sets the international poverty line at $2.15 per day per person, the lower middle-income class poverty line at $3.65, and the upper middle-income class poverty line at $6.85 [46]. According to the World Bank [47], 9.3% of Senegal's population lives below the international poverty line, with 37.4% below the lower middle-income class poverty line and 74.4% below the upper middle-income class poverty line.

Between 2018 and 2022, the percentage of households below the international poverty line slightly increased from 3.33% to 3.95%. Meanwhile, households below the lower middle-income poverty line increased from 18.33% in 2018 to 29.38% in 2022, and those below the upper middle-income poverty line increased from 58.33% to 76.84%. These results indicate an overall decline in daily household consumption over the four years.

## Results

### Preliminary analysis

Based on the aggregated standardized BW scores from the 2018 and 2022 surveys, we compared the responses for each value item. Fig 5 displays the mean standardized scores for each value item. A positive score indicates that respondents consider the value important, while a negative score suggests the opposite. This comparison highlights respondents' value tendencies over time. In both surveys, tradition—representing family and religious values—consistently ranked as the most important. Respondents also emphasized benevolence-caring, symbolizing kindness toward others, and conformity, reflecting social discipline. These values correspond to the conservation and self-transcendence dimensions of values.

However, significant changes were observed in the standardized BW scores for values such as tradition and benevolence over the past four years. Values that were consistently viewed as less important in both 2018 and 2022 included hedonism and stimulation, both of which belong to the openness to change dimension. In particular, hedonism had lower scores even before the pandemic, and these values decreased further post-pandemic.

In addition, values such as universalism (concern for environmental protection) and self-direction (emphasizing creativity), which were ranked low but still preferred in 2018, became the least preferred in 2022. In contrast, values in the self-enhancement category, including power (representing economic success) and achievement (representing social success), which were the least favored in 2018, saw a significant increase in scores by 2022. This shift indicates a more significant contrast in value preferences between the pre-pandemic and post-pandemic periods.

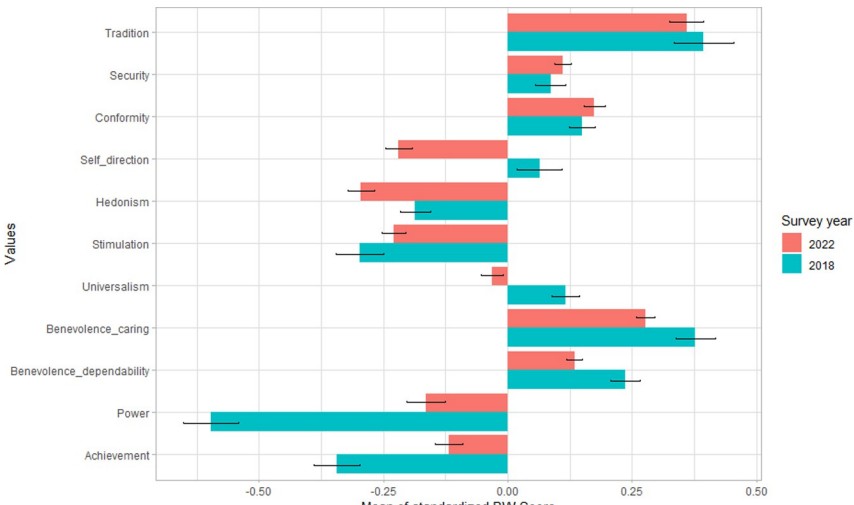

**Fig 5. Standardized BW scores in 2018 and 2022 surveys.** Note: Value items are ranked by relative importance based on standardized BW scores. Positive scores indicate higher importance, while negative scores represent lower importance. Error bars represent the standard error of the mean.

## Change in values before and after the COVID-19 Pandemic

Table 4 presents the results of statistical tests conducted to evaluate the differences in the mean values for each item between 2018 and 2022. Given the unequal sample sizes in the two surveys, a standard t-test assuming equal variances was not applicable. Instead, the Welch test was employed to account for these differences. A two-tailed test was conducted with the null hypothesis that there were no significant changes in the values of respondents between 2018 and 2022. If a significant difference in the mean values was found, the null hypothesis was rejected.

**Table 4. Results of Welch's t-test.**

|  | MEAN_2018 | MEAN_2022 | Differences | SE | *t*-value | *p*-value | CI95_L | CI95_U |
|---|---|---|---|---|---|---|---|---|
| *Conservation* | | | | | | | | |
| Tradition | 0.394 | 0.360 | -0.034 | 0.070 | -0.495 | 0.622 | -0.173 | 0.104 |
| Security | 0.086 | 0.111 | 0.025 | 0.035 | 0.713 | 0.477 | -0.044 | 0.094 |
| Conformity | 0.150 | 0.174 | 0.024 | 0.033 | 0.726 | 0.469 | -0.042 | 0.090 |
| *Openness to change* | | | | | | | | |
| Self-direction | 0.064 | -0.219 | -0.283 | 0.052 | -5.420 | 0.000*** | -0.387 | -0.180 |
| Hedonism | -0.186 | -0.295 | -0.109 | 0.041 | -2.658 | 0.009** | -0.190 | -0.028 |
| Stimulation | -0.297 | -0.229 | 0.068 | 0.054 | 1.260 | 0.211 | -0.039 | 0.175 |
| *Self-Transcendence* | | | | | | | | |
| Universalism | 0.117 | -0.032 | -0.148 | 0.036 | -4.158 | 0.000*** | -0.219 | -0.078 |
| Benevolence-caring | 0.378 | 0.277 | -0.100 | 0.044 | -2.269 | 0.026* | -0.188 | -0.012 |
| Benevolence-dependability | 0.236 | 0.134 | -0.102 | 0.033 | -3.044 | 0.003** | -0.168 | -0.035 |
| *Self-Enhancement* | | | | | | | | |
| Power | -0.597 | -0.164 | 0.433 | 0.068 | 6.400 | 0.000*** | 0.299 | 0.567 |
| Achievement | -0.344 | -0.117 | 0.227 | 0.054 | 4.217 | 0.000*** | 0.120 | 0.334 |

Note:*, **, *** indicate that the coefficients are statistically different from zero at the 10, 5, and 1 percent level, respectively.

The results, including the means, standard errors, differences in means, test statistics, and *p*-values, are summarized in Table 4. The Welch test revealed significant differences in value items related to self-transcendence (including universalism, benevolence-caring, and benevolence-dependability), openness to change (represented by self-direction and hedonism), and self-enhancement (represented by power and achievement). The following sections provide further discussion of the changes in these four value categories.

## Examination of each value groups

### Conservation values

Conservation emphasizes social order, stability, and traditional customs. The changes in scores for conservation values were as follows: tradition (-0.034), security (+0.025), and conformity (+0.024). Welch's t-test results indicated no significant changes due to the pandemic: tradition (t = -0.495, *p*<0.622), security (t = 0.713, *p*<0.477), and conformity (t = 0.726, *p*<0.469).

### Openness to change values

Openness to change represents an appreciation for creativity and personal enjoyment. Changes in mean scores were observed in self-direction (-0.283), stimulation (+0.068), and hedonism (-0.109), with self-direction showing a significant decrease. The Welch test confirmed significant differences for self-direction (t = -5.420, *p*<0.000) and hedonism (t = -2.658, *p*<0.009). These values, which were not considered important in 2018, became even less important in 2022.

### Self-transcendence values

Benevolence-dependability (-0.102), benevolence-caring (-0.100), and universalism (-0.148) showed decreased scores. Despite its relatively low score, universalism, which pertains to environmental protection, was still emphasized in 2018. However, this was no longer the case by 2022. Welch's t-test results revealed significant differences for benevolence-dependability (t = -3.044, *p*<0.003), benevolence-caring (t = -2.269, *p*<0.026), and universalism (t = -4.158, *p*<0.000).

### Self-enhancement values

Both power (t = 6.400, *p*<0.000) and achievement (t = 4.217, *p*<0.000), which were not considered important in 2018, showed significant differences in 2022. Scores for power (+0.433) and achievement (+0.227) increased, indicating a shift toward valuing self-interest, financial success, and social status.

### Summary

Using the Welch test, we examined the changes in values and identified those with significant differences. The overall trend in values remained consistent across 2018 and 2022. Tradition (from the conservation group), along with benevolence-caring and benevolence-dependability (from the self-transcendence group), were identified as the most important values in both years. However, achievement and power (self-enhancement values), which were not deemed important in 2018, became more significant in 2022. Conversely, the importance of hedonism and self-direction (openness to change values) declined significantly.

The examination of changes in standardized BW scores across the two time points revealed several values with significant differences. Power and achievement, which reflect financial success and social status, showed a significant increase in scores. In contrast, self-direction and

hedonism, values associated with creativity and personal enjoyment, experienced a decline. Additionally, scores for benevolence-caring, benevolence-dependability, and universalism—values tied to environmental and societal contributions—also decreased significantly.

## Discussion

This study aimed to quantitatively assess changes in the values of urban residents in Senegal during a global crisis. Previous research on values has demonstrated that cultural similarities often lead to common values tendencies [24,25]. To validate this survey, we compared its results in Senegal with the WVS conducted in Mali and Burkina Faso, both member countries of the Economic Community of West African States (ECOWAS)(Note2). This comparison helped us assess similarities, differences, and the robustness of the findings.

Table 5 presents the ranking of value items from the WVS in Mali and Burkina Faso [48], comparing them with our survey results from Senegal. Since the Senegalese respondents were female, we also restricted the analysis of the Mali and Burkina Faso surveys to female respondents for consistency. Our survey included two dimensions, benevolence-caring and benevolence-dependability, covering a total of 11 value items. In contrast, the WVS surveys in Burkina Faso and Mali focused on the broader benevolence dimension, with 10 value items. Despite these slight differences, it was still possible to compare the values expressed across these surveys.

As shown in Table 5, the values of benevolence under self-transcendence, and tradition and conformity under conservation, were consistently prioritized across the three countries. On the contrary, values such as stimulation, hedonism, and power were not highly preferred. This indicates a common trend in Senegal and other West African countries, where self-transcendence values (such as benevolence) and conservation values (such as tradition and social cohesion) are prioritized, while self-enhancement and openness to change values (such as financial success and hedonism) are less emphasized. These shared value patterns across West African countries may reflect common cultural and social factors, lending confidence to the results of this survey given the observed similarities.

**Table 5. Comparison of values in francophone West African countries.**

| Dimension | Value attributions | Mali (N = 601) | Burkina Faso (N = 579) | Senegal 2018 (N = 60) | Senegal 2022 (N = 200) |
|---|---|---|---|---|---|
| *Conservation* | Tradition | 3 | 3 | 1 | 1 |
| | Security | 2 | 1 | 6 | 5 |
| | Conformity | 4 | 4 | 4 | 3 |
| *Openness to Change* | Self-direction | 7 | 6 | 7 | 9 |
| | Stimulation | 9 | 8 | 9 | 10 |
| | Hedonism | 10 | 10 | 8 | 11 |
| *Self-Transcendence* | Benevolence-caring[a] | | | 2 | 2 |
| | Benevolence-dependability [b] | 1 | 2 | 3 | 4 |
| | Universalism | 5 | 7 | 5 | 6 |
| *Self-Enhancement* | Power | 6 | 5 | 10 | 7 |
| | Achievement | 8 | 9 | 11 | 8 |

Note: Mali and Burkina Faso data were from the World Values Survey[48], while the Senegal data were from the survey. In the values survey conducted in Mali and Burkina Faso, respondents' preferred values were assessed using a 6-point scale ranging from Very much like me or Not at all like me. In the survey in Senegal, Best Worst Scaling results were used to calculate respondents' value preference scores. Benevolence attribution can be divided into Benevolence-caring[a] and Benevolence-dependability[b].

Numerous studies have shown that when people feel anxious about risks and threats, they tend to emphasize values related to self-preservation, such as tradition and security. Conversely, values associated with openness to change, such as stimulation and self-direction, often decrease as people adapt to new situations [13–15]. In this study, a comparison of 2018 and 2022 data revealed no significant changes in conservation values. This indicates that value changes among urban Senegalese people deviate from the trends observed in previous research. Specifically, during the global crisis, urban Senegalese did not significantly increase their conservative values. However, self-direction and hedonism—values related to openness to change—decreased significantly.

A previous study in Australia found that when people experienced long-term behavioral restrictions and fewer opportunities for social interaction, openness to change values such as self-direction and stimulation diminished in importance. Subsequently, values related to self-transcendence, specifically benevolence and universalism, also declined, leading to heightened individual anxiety [17]. However, cultural differences in infection control measures have been noted [11], highlighting the need to consider individual cultural contexts when examining changes in values.

In Senegal, anxiety related to the risk of COVID-19 infection was relatively low (36%) compared to neighboring West African countries [12]. Supporting this, the percentage of respondents in this survey who invited friends, relatives, and others for meals—a sign of social interaction—increased after the pandemic compared to before it (Table 3). These results indicate that social distancing in Senegal was less rigid than anticipated. Therefore, the decline in values related to openness to change, such as self-direction and hedonism, may not stem from reduced social interaction. Instead, it appears to reflect a social context where economic stagnation makes it challenging for individuals to prioritize personal interests.

Values associated with self-transcendence, such as benevolence-dependability, benevolence-caring, and universalism, also declined. These values reflect a focus on transcending individual concerns to contribute to others and show compassion for society. Previous studies [16, 17] have shown that during a pandemic, personal values tend to decrease as people are forced to prioritize their own health and safety over societal concerns. The marked decrease in universalism, which reflects a concern for the natural environment, is particularly notable and aligns with Daniel et al.'s [17] findings that interest in environmental protection waned during the pandemic.

Conversely, power and achievement—values under self-enhancement—were among the least significant in 2018, but their BW scores increased significantly by 2022. Previous studies on value changes among European youth during the financial crisis revealed divergent value preferences based on countries' investments in social security. In countries with low social security investment, values such as power and achievement increased during the global financial crisis, whereas values such as tradition, benevolence, and security declined. In contrast, countries with higher social security investment saw an increase in the favorability of tradition, benevolence, and conformity [14]. This suggests that generous social security policies may contribute to lower anxiety levels. Given the rising importance of power and achievement in Senegal, this trend appears similar to that of countries with lower social security investment.

According to the World Social Protection report [49], the African region experienced a 7.7% reduction in working hours due to the economic crisis following COVID-19, equivalent to the loss of 29 million jobs. However, in West Africa, unemployment compensation and wage subsidies covered only 4.6% of the unemployed population, significantly below the global average of 18.6%. With 85.8% of African workers classified as informal workers, few are eligible for these benefits. In addition, West African workers receive minimal social security

support, including childcare allowances (11.5%) and maternity benefits (6.8%). These statistics highlight the low investment in social security across West African countries.

While the economy has yet to recover from the economic stagnation triggered by the pandemic, the rising cost of living, exacerbated by the Russia-Ukraine war, has further worsened living conditions, particularly in low- and middle-income countries. Various international aid programs have been implemented in the social security sector since 2019 [50]. Individual values likely reflect the long-standing fragility of social security systems in Africa.

A food price report released by the Senegalese government [51] revealed that the prices of essential items such as wheat, fertilizer, and fuel have surged, especially since the onset of the Russia-Ukraine war. Interviews conducted between June and July 2022 indicated that 51% of respondents attributed rising food prices to the war, 6% to trade stagnation caused by the pandemic, and 16% to a combination of the war, pandemic, and other factors. In Africa, large extended families are common, with many households supporting children of relatives who have moved to urban centers for education. Therefore, the rapid increase in food prices poses a serious challenge for large families. Respondents emphasized that these rising prices directly impact their ability to make ends meet, forcing many households to adopt coping strategies such as skipping dinner or switching to more affordable foods. According to the world Bank, this shift toward cheaper meals is a significant trend in Senegal [47], aligning with our findings.

Additionally, some households reported that the economic stagnation following the COVID-19 pandemic forced family members to take significant risks, such as attempting perilous sea crossings in small boats to seek job opportunities in Europe. Further fiscal stimulus is needed to address pandemic-related challenges and strengthen social security. Complex issues, including rising food prices and increasing unemployment, have caused substantial financial damage. Per capita consumption expenditure decreased by 19.9% in 2022 compared to 2018. The 2022 survey indicated a distressing trend: 29.38% of households were living on less than $3.65 per person per day, an increase of 11.05% from 2018 (Table 3). These households fall within the lower-middle income bracket and endure harsh living conditions. Such circumstances are likely to significantly impact the values and perspectives of the Senegalese people.

To fully evaluate the changes in values discussed in this study, ongoing observation is necessary to determine whether these shifts are temporary or permanent. Since this study compares two distinct periods, it is difficult to assess the permanence of the change observed. Previous research has identified three main categories of value change [52]. First, short-term changes occur due to psychological experiments [53]. Second, intra-individual value changes are triggered by external social events such as war, economic crises, immigration, and pandemics [13, 14, 29, 54], as well as personal life events such as schooling, marriage, parenthood, and retirement [19, 26, 55]. Third, value changes are associated with aging [56]. Although individual values can shift significantly in response to major events such as terrorism, financial crises, or immigration, they tend to revert to their original state over time [14, 15, 54]. While these changes can be seen as adaptations to new social challenges or stressors [54], the direction of value shifts depends on an individual's inherent values and cultural context.

Intra-individual value change associated with aging involves a gradual shift in values over the course of a lifetime [22]. For instance, as people age and experience life events, they tend to place greater importance on conservation and self-transcendence, while openness to change diminishes [56]. However, recent studies suggest that value differences also vary by generation. These generational differences include both value changes over time and internal changes within individuals. These shifts may have long-term implications for the development of social values [52]. It is conceivable that a temporary change in values caused by a major social event could influence the value formation of a particular generation.

Furthermore, a generation that has undergone such a shift may continue to change over time due to aging and life experiences, resulting in differences compared to other generations. However, most of these studies have been limited to wealthier countries and need to be expanded. It is imperative to further investigate whether the value changes observed in this study are temporary or lasting, considering factors such as regional and cultural differences, personal experiences with global crises, and socioeconomic conditions.

## Conclusions

Human behavior reflects the value-oriented culture and norms of the society to which individuals belong. Understanding how significant events, such as the COVID-19 pandemic and international conflicts, influence value changes in Africa is essential. This study explores the impact of these events on individual values, as well as the cultural norms and customs that shape behavior among African urban dwellers.

Previous research indicates that during crises, people tend to place greater emphasis on conservative values and focus more on social cohesion. However, this study found a different trend among respondents in Senegal. Contrary to expectations, there was no increase in the importance of conservation values typically reported in previous studies [7, 16]. Instead, urban residents in Senegal gave more importance to values such as power and achievement in 2022—values that were considered least important in 2018. Conversely, values related to self-transcendence, such as universalism, benevolence-caring, and benevolence-dependability, which have traditionally been highly regarded in the West African region, decreased in significance after the pandemic. In addition, the importance of self-direction and hedonism also declined (Table 4).

Various factors, including war, global financial crises, pandemic-induced behavioral constraints, social security systems, and economic conditions, have been shown to influence individual values [7, 13, 14, 16, 17]. This study suggests that not only pandemic-related issues, such as movement restrictions and social distancing but also rising unemployment and inflation, combined with weak social security systems, played a major role in shifting individual values, particularly in Senegal.

While the world recovers from the stagnation caused by the COVID-19 pandemic, the Russia-Ukraine war is ongoing. Consequently, people in low- and middle-income countries, including those in Africa, are expected to continue facing difficulties in accessing adequate food, as energy and food prices remain high [57]. Africa is also experiencing rapid urbanization, with many people transitioning from traditional rural livelihoods to urban lifestyles [57, 58]. By 2030, over 50% of Senegal's population is projected to be living in urban areas [59]. These long-term changes are likely to reshape the social structure, further influencing individual values in the years to come.

Rapid urbanization during the COVID-19 pandemic has driven the expansion of the mobile phone market [60] and brought financial shifts through the spread of mobile payments [61]. However, external shocks such as the pandemic and the Russia-Ukraine war have also contributed to high levels of informal employment and inflation in Senegal. These social changes are expected to have a significant cultural impact. The educational disparity between the two groups in this study suggests that differences in education may be linked to the observed changes in values. Differences in how individuals respond to crises may result from variations in their ability to access information, participate in COVID-19 sanitation efforts, and seize work opportunities.

However, when comparing years of education and employment rates by district in the 2022 survey, respondents in districts closer to commercial areas showed higher employment rates,

suggesting that education level may not directly correlate with work opportunities. In addition, while public education in Senegal is provided in French, media and social networking sites offer information in the widely spoken Wolof language, allowing even the illiterate population to access a broad range of information. Therefore, the impact of educational differences on value changes appears to be minimal at this time. Nonetheless, it is too early to draw firm conclusions about how cultural shifts are affecting values; more detailed data and further analysis of previous studies from multiple perspectives are needed.

The global crisis has exposed the inadequacy of Senegal's social security system in adapting to a rapidly changing environment and underscored the vulnerability of human life to various risks. In response to crises such as COVID-19 and the Russia-Ukraine war, there may have been a shift in values, with individuals prioritizing their personal interests over the collective good. In addition, as many African regions, including Senegal, experience population growth and urbanization, they are becoming increasingly integrated into the global context, making them more susceptible to the impacts of global crises. Given these circumstances, there is a pressing need to develop robust health and social security systems, particularly in light of the high probability that mutual trust and cooperation may be destabilized during future crises.

## Limitations

Although the present study reveals important findings, it is important to acknowledge its limitations. First, this study focuses specifically on Saint-Louis, Senegal, limiting its external validity. Future studies should examine whether these findings apply to other regions. Furthermore, due to financial limitations, the sample size is relatively modest compared to larger surveys conducted by international organizations. As a result, the study may not have fully captured the extent of the changes that occurred. Furthermore, the selection methods for the first and second surveys differed, which may affect the comparability of the data.

This study presents a straightforward comparison between two distinct time points, so the impacts of global crises, such as the COVID-19 pandemic and the Russia-Ukraine conflict, are difficult to isolate. It is not possible to identify the individual effects of the various factors that occurred during this period; they can only be presented as hypotheses.

Although a strong emphasis on social contribution, traditions, and religious norms is common in francophone West African countries, long-term observation is necessary to determine whether urbanization may alter these trends in the future. Additionally, it would be worthwhile to explore whether the value items used in this study are applicable for comparison with urban residents in other African countries. From this perspective, the study of Senegalese urban residents offers novel insights into how major social changes may influence future value shifts.

Note 1: The two surveys were designed to collect data on household attributes, food consumption, and values over a continuous five-day period. This study focuses specifically on the values survey.

Note 2: The Economic Community of West African States (ECOWAS) consists of 15 member countries: Benin, Burkina Faso, Cape Verde, Côte d'Ivoire, The Gambia, Ghana, Guinea, Guinea-Bissau, Liberia, Mali, Niger, Nigeria, Senegal, Sierra Leone, and Togo.

## Supporting information

**S1 file. Data set for respondents, and txt for generating tables and figures.**
(ZIP)

## Acknowledgments

We express our heartfelt gratitude to the dedicated students from Gaston Berger University, for their diligent efforts in conducting the survey. Furthermore, we extend our appreciation to Dr. Hideo Aizaki from Hokkaido University, Dr. Papa Serr Saliou and Dr. Baltazar Antonio from JIRCAS, Nobuyuki Nishiyama from Earth and Human corporation, Masahiko Taniguchi from JA Tourism & Communications, Maimouna Ndour from Africa Rice Center, Dr. Farokh Niass from Gaston Berger University for their valuable comments, insightful suggestions, and unwavering support throughout this research endeavor. We thanks to kind support by The Japan Society for the Promotion of Science, Overseas Challenge Program for Young Researchers. Lastly, we would like to convey our sincere thanks to the households that generously participated in this study, dedicating their time and sharing invaluable information that significantly contributed to the success of our research. Our sincere condolences go to co-author Dr. Moussa Ndong, who passed away in COVID-19 in 2021.

## Author Contributions

**Conceptualization:** Yachiyo Tobita, Kiyokazu Ujiie.

**Data curation:** Yachiyo Tobita.

**Formal analysis:** Yachiyo Tobita, Kiyokazu Ujiie.

**Funding acquisition:** Yachiyo Tobita, Kiyokazu Ujiie.

**Investigation:** Yachiyo Tobita, Mandiaye Diagne, Joseph Bassama, Moussa Ndong, Mor Gueye, Kiyokazu Ujiie.

**Methodology:** Yachiyo Tobita, Kiyokazu Ujiie.

**Resources:** Mandiaye Diagne.

**Supervision:** Mandiaye Diagne, Kiyokazu Ujiie.

**Visualization:** Yachiyo Tobita, Kiyokazu Ujiie.

**Writing – original draft:** Yachiyo Tobita, Kiyokazu Ujiie.

**Writing – review & editing:** Yachiyo Tobita, Mandiaye Diagne, Joseph Bassama, Mor Gueye, Kiyokazu Ujiie.

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
