## [Decision Letter · Decision Letter 0]

10 Apr 2024

PONE-D-23-37707Study assesses changes in values of urban Senegalese during global crisesPLOS ONE

Dear Dr. Tobita,

Thank you for submitting your manuscript to PLOS ONE. After careful consideration, we feel that it has merit but does not fully meet PLOS ONE’s publication criteria as it currently stands. Therefore, we invite you to submit a revised version of the manuscript that addresses the points raised during the review process.

We look forward to receiving your revised manuscript.

Kind regards,

Hamed Ahmadinia

Academic Editor

PLOS ONE

2. You indicated that ethical approval was not necessary for initial 2018 study. Could you please provide confirmation from your institutional review board or research ethics committee (e.g., in the form of a letter or email correspondence) that ethics review was not necessary for this study? Please include a copy of the correspondence as an ""Other"" file.

Additionally please include a complete copy of PLOS’ questionnaire on inclusivity in global research in your revised manuscript. Our policy for research in this area aims to improve transparency in the reporting of research performed outside of researchers’ own country or community. The policy applies to researchers who have travelled to a different country to conduct research, research with Indigenous populations or their lands, and research on cultural artefacts. The questionnaire can also be requested at the journal’s discretion for any other submissions, even if these conditions are not met.  Please find more information on the policy and a link to download a blank copy of the questionnaire here: https://journals.plos.org/plosone/s/best-practices-in-research-reporting. Please upload a completed version of your questionnaire as Supporting Information when you resubmit your manuscript.

6. Please upload a copy of Figure 1 and 2, to which you refer in your text on page 13 and 20. If the figure is no longer to be included as part of the submission please remove all reference to it within the text.

7. Please upload a copy of Table S1, S2, S3 and S4 which you refer to in your text on page 38.  Please upload them with the file type 'Supporting Information'. Please ensure that each Supporting Information file has a legend listed in the manuscript after the references list.  

8. All supplementary figures and tables are must be uploaded with the file type 'Figure'. Please amend the file type to 'Supporting Information'. Please ensure that each Supporting Information file has a legend listed in the manuscript after the references list.

Additional Editor Comments:

Dear Authors,

I have received the review of your manuscript, as provided by our esteemed reviewer. The submission, which focuses on the changes in values of urban Senegalese during significant global crises, presents important insights that contribute to the field's understanding of societal value shifts in response to events like the COVID-19 pandemic and the Russia-Ukraine War. Your utilization of Schwartz's ten value dimensions and the innovative approach using Best-Worst Scaling (BWS) adds notable depth to this area of study.

However, before your manuscript can be considered for publication in PLOS ONE, some issues, as highlighted by the reviewer, require your attention. Addressing these points will not only enhance the clarity and depth of your research but also expand its relevance and applicability:

Broader Implications: While your study adeptly addresses the immediate research questions, it is necessary to extend the discussion to include broader implications. Specifically, a deeper exploration of how these insights might be utilized by policymakers, health professionals, and other stakeholders is vital. This will enhance the practical utility of your findings and underscore their relevance in the current global context.

Temporary vs. Persistent Value Changes: The concept of whether value changes are temporary or persistent is intriguing and deserves more thorough examination. Your paper would benefit from a detailed discussion on the significance of both types of changes, offering insight into their respective implications within the context of your study.

Sampling Size and Potential Biases: The disparity in sample sizes between your 2018 and 2022 surveys raises questions regarding comparability and potential biases. A more detailed explanation of how these differences were addressed in your analysis, as well as how the results can be generalized to the broader urban population in Senegal, is crucial for the credibility and reliability of your findings.

Consideration of Other External Factors: Your study currently focuses on the period between pre- and post-pandemic. However, it is important to consider other external variables that might have influenced societal value shifts during this timeframe. Factors such as cultural transformations, educational differences, and varying sample sizes should be thoroughly discussed to provide a more comprehensive understanding of the observed changes.

Your research contributes significantly to our understanding of societal dynamics during crises, and addressing these concerns will strengthen your findings and broaden the scope of your study. Therefore, I suggest that the manuscript be revised according to the above recommendations. Once these minor revisions are implemented, your paper will be better positioned for publication in PLOS ONE.

I look forward to receiving your revised manuscript and believe that the suggested changes will greatly enhance the value and impact of your work.

Sincerely,

Hamed Ahmadinia

Academic Editor

PLOS ONE

Reviewers' comments:

Reviewer's Responses to Questions

**Comments to the Author**

1. Is the manuscript technically sound, and do the data support the conclusions?

Reviewer #1: Yes

2. Has the statistical analysis been performed appropriately and rigorously? 

Reviewer #1: Yes

3. Have the authors made all data underlying the findings in their manuscript fully available?

Reviewer #1: Yes

4. Is the manuscript presented in an intelligible fashion and written in standard English?

Reviewer #1: Yes

5. Review Comments to the Author

Reviewer #1: Dear author,

The research on “Study assesses changes in values of urban Senegalese during global crises” presents valuable insights into the dynamic nature of societal values in response to crises such as the COVID-19 pandemic and the Russia-Ukraine War. Its application of Schwartz's ten value dimensions and the innovative use of Best-Worst Scaling (BWS) contribute significantly to understanding these shifts. However, I would like to address a few areas that require further clarification or expansion:

The relevance of research in the current context is significant. Your study provides a crucial understanding of societal shifts during crises, which is highly relevant given the current global situation. While the research aligns well with the need for understanding societal dynamics in times of crisis, it would be beneficial to further articulate the broader implications of these findings. Specifically, how might these insights be utilised by policymakers, health professionals, or other stakeholders to better prepare for and respond to future crises?

Temporary vs. persistent value changes: Your paper briefly mentions that individual values may revert to their original state over time. This point is particularly intriguing and warrants further exploration. Could you elaborate on the implications of temporary versus persistent value changes in your study? Are temporary value changes equally significant or is the focus just on enduring changes? Could you elaborate on the impacts of temporary value changes?

Sampling size and potential biases: The disparity in sample sizes between the 2018 (60) and 2022 (200) surveys raises concerns about comparability and potential biases. The larger sample size in 2022 might lead to more nuanced insights and analyses that were not feasible with the smaller sample in 2018. How you have addressed these sampling differences in your analysis? and how You generalise your results to the broader urban population in Senegal?

Consideration of Other External Factors: This study captures the societal value shifts between the pre- and post-pandemic periods. However, it is critical to consider other intervening variables that may have influenced these changes across the four-year study period. For example, cultural transformations, educational differences, and sample size differences, could all have a significant impact. Could you elaborate on how your study considered these factors?

Sincerely

Reviewer

6. PLOS authors have the option to publish the peer review history of their article (what does this mean?). If published, this will include your full peer review and any attached files.

Reviewer #1: **Yes: **

---

## [Author Response · Author response to Decision Letter 0]

15 Jun 2024

Point-by-point Response to the Editor and Reviewer

Journal requirements

1. When submitting your revision, we need you to address these additional requirements. Please ensure that your manuscript meets PLOS ONE’s style requirements, including those for file naming. 

RESPONSE: We have changed the style to align it with the specifications of the PLOS ONE style.

2. You indicated that ethical approval was not necessary for initial 2018 study. Could you please provide confirmation from your institutional review board or research ethics committee (e.g., in the form of a letter or email correspondence) that ethics review was not necessary for this study? Please include a copy of the correspondence as an “Other” file. Additionally please include a complete copy of PLOS’ questionnaire on inclusivity in global research in your revised manuscript. Our policy for research in this area aims to improve transparency in the reporting of research performed outside of researchers’ own country or community. The policy applies to researchers who have travelled to a different country to conduct research, research with Indigenous populations or their lands, and research on cultural artefacts. The questionnaire can also be requested at the journal’s discretion for any other submissions, even if these conditions are not met. Please find more information on the policy and a link to download a blank copy of the questionnaire here: https://journals.plos.org/plosone/s/best-practices-in-research-reporting.Please upload a completed version of your questionnaire as Supporting Information when you resubmit your manuscript.

RESPONSE: We apologize for the confusion caused by the lack of explanation regarding the ethical approval for the 2018 survey. We submitted all the necessary information, including the methodology and questionnaire, to the Senegalese Ministry of Higher Education and Innovation, which has an ethics review committee, as Gaston Berge University does not have an ethics committee. We obtained a research permit, including a comprehensive ethical approval, for the collaborative research project with Gaston Berger University. According to co-author Dr. Moussa Ndong, ethics approval is a part of the research approval process and serves as an official certification. We kindly ask you to verify the research permit, including a comprehensive ethical approval, letter issued by the Ministry of Higher Education and Innovation in 2018, Senegal, which we have already submitted as “Autorisation_Rechercher_2018. pdf”. 

We have uploaded the PLOS’ questionnaire on inclusivity in global research as Supporting Information and reported in section Method. 

RESPONSE: Thank you for pointing this out. The grant organization, the Japan Society for the Promotion of Science, Overseas Challenge Program for Young Researcher indicated that only the support should be mentioned in the acknowledgements. Therefore, we changed our grant information as follow;

The study in 2018 was supported by public interest incorporated foundation, Urakami Foundation for Food and Food Culture Promotion (https://www.urakamizaidan.or.jp/en.html), grant number: 2017-30 (K.U.) and 

public interest incorporated foundation, Food Culture Promotion and Ajinomoto Foundation for Dietary Culture (https://www.syokubunka.or.jp/english/),　grant number: 2017-01(Y.T. and K.U.). The study in 2022 was supported by ST-SPRING, Japan Science and Technology Agency (https://www.jst.go.jp/EN/),　grant number :JPMJSP2124(Y.T.). The funders had no role in study design, data collection and analysis, decision to publish, or preparation of the manuscript.

4. We note that your Data Availability Statement is currently as follows: [All relevant data are within the manuscript and its Supporting Information files.] Please confirm at this time whether or not your submission contains all raw data required to replicate the results of your study. Authors must share the “minimal data set” for their submission. PLOS defines the minimal data set to consist of the data required to replicate all study findings reported in the article, as well as related metadata and methods (https://journals.plos.org/plosone/s/data-availability#loc-minimal-data-set-definition).

RESPONSE: Thank you for clarifying the requirements regarding the submission of the minimal dataset. The Ethics Committee of the University of Tsukuba prescribes that only statistical data can be provided when submitting papers or presenting at conferences. Therefore, we have uploaded the following statistical data and R script of Table 2, Table 3, and Figure 3 in the Supporting Information. We also changed the Data Availability Statement as follows;

Data Availability Statement : The relevant data and R script for analysis and visualization are available in the Support Information. The open-source datasets used in Table 4 is available at: WVS Database (worldvaluessurvey.org). 

RESPONSE: Thank you for the comment. We have added the full ethics statement and obtained informed written consent in Method section (lines 283 to 287)

[Written consent was obtained from before the survey began. The comprehensive research permission including ethical approval for the 2018 survey was obtained from the Ministry of Higher Education Research and Innovation in Senegal (Approval No:0000303), and ethical approval for the 2022 survey was obtained from the Faculty of Life and Environmental Sciences of the University of Tsukuba(No:2022-2).]

6. Please upload a copy of Figure 1 and 2, to which you refer in your text on page 13 and 20. If the figure is no longer to be included as part of the submission please remove all reference to it within the text.

RESPONSE: We have uploaded Figure 1 and Figure 2. 

7. Please upload a copy of Table S1, S2, S3 and S4 which you refer to in your text on page 38. Please upload them with the file type ‘Supporting Information’. Please ensure that each Supporting Information file has a legend listed in the manuscript after the references list. 

RESPONSE: We have uploaded S1 file.(zip) as dataset and R script and added it in the manuscript after the reference list. Kindly refer our response about data availability in the journal requirement 4. 

8. All supplementary figures and tables must be uploaded with the file type ‘Figure’. Please amend the file type to ‘Supporting Information’. Please ensure that each Supporting Information file has a legend listed in the manuscript after the references.

RESPONSE: We have no supplemental tables or figures. 

RESPONSE: We sincerely appreciate your comments. We have thoroughly reviewed our reference list to ensure its completeness and accuracy. To expand our discussion, we are adding 12 references to the revised manuscript as follow;

1. World Value Survey, Inglehart, R., C. Haerpfer, A. Moreno, C. Welzel, K. Kizilova, J. Diez-Medrano et al. (eds.). 2014. World Values Survey: Round Six - Country-Pooled Datafile Version 

2. ANSD (Agence National de la Statistiqué et de la Démographie, Senegal) L’EHCVM est une des composantes principales du programme d’harmonisation et de modernisation des enquêtes sur les conditions de vie. 

3. ANSD (Agence National de la Statistiqué et de la Démographie, Senegal) Projet a l’ecoute du Senegal 2014.

4. World Bank, Poverty & Equality Brief, Senegal 2022. Available from: Poverty and Equity Briefs (worldbank.org)

5. Inglehart, R., C. Haerpfer, A. Moreno, C. Welzel, K. Kizilova, J. Diez-Medrano et al. (eds.). 2014. World Values Survey: All Rounds - Country-Pooled Datafile Version.

6. Leijen, I., Van Herk, H., & Bardi, A. (2022). Individual and generational value change in an adult population, a 12-year longitudinal panel study. Scientific Reports.

7. Arieli, S., Grant, A. M., & Sagiv, L. (2014). Convincing Yourself to Care About Others: An Intervention for Enhancing Benevolence Values. Journal of Personality.

8. Lönnqvist, J.-E., Jasinskaja-Lahti, I., & Verkasalo, M. (2011). Personal Values Before and After Migration. Social Psychological and Personality Science.

9. Lönnqvist, J., Leikas, S., & Verkasalo, M. (2017). Value change in men and women entering parenthood: New mothers' value priorities shift towards Conservation values. Personality and Individual Differences.

10. Milfont, T. L., Milojev, P., & Sibley, C. G. (2016). Values Stability and Change in Adulthood. Personality and Social Psychology Bulletin.

11. African Business 2020, Cover story: Pandemic impact accelerates long-term trends for Orange.

12. GSMA The Mobile Economy Sub-Sahara Africa 2020.

Response to additional Editor Comments:

1: Broader Implications: While your study adeptly addresses the immediate research questions, it is necessary to extend the discussion to include broader implications. Specifically, a deeper exploration of how these insights might be utilized by policymakers, health professionals, and other stakeholders is vital. This will enhance the practical utility of your findings and underscore their relevance in the current global context.

RESPONSE: Thank you for this valuable comment. We have extended the discussion to include the broader implications of our study and provide insights into how it might be utilized by policymakers, health professionals, and other stakeholders. Details have been included in our responses to the reviewer’s comment.

2: Temporary vs. Persistent Value Changes: The concept of whether value changes are temporary or persistent is intriguing and deserves more thorough examination. Your paper would benefit from a detailed discussion on the significance of both types of changes, offering insight into their respective implications within the context of your study.

RESPONSE: Thank you for this insightful comment. We have provided a detailed discussion of temporary and persistent value changes, highlighting the significance of both types of changes and their respective implications within the context of our study. Details have been included in our responses to the reviewer’s comment.

3: Sampling Size and Potential Biases: The disparity in sample sizes between your 2018 and 2022 surveys raises questions regarding comparability and potential biases. A more detailed explanation of how these differences were addressed in your analysis, as well as how the results can be generalized to the broader urban population in Senegal, is crucial for the credibility and reliability of your findings.

REPONSE: We have explained how we addressed issues of comparability and potential bias with respect to differences in sample size. Details have been included in our responses to the reviewer’s comment.

4:Consideration of Other External Factors: Your study currently focuses on the period between pre- and post-pandemic. However, it is important to consider other external variables that might have influenced societal value shifts during this timeframe. Factors such as cultural transformations, educational differences, and varying sample sizes should be thoroughly discussed to provide a more comprehensive understanding of the observed changes.

RESPONSE: Thank you for the suggestion. In the revised version, we have explored the external variables that may have influenced social value shifts between the pre- and post-pandemic periods. We discuss how the global crisis highlighted the inadequacy of the Senegalese social security system in the face of a dynamic social environment and the vulnerability of human life to various risks. Regarding educational differences, we explain how the differences in the number of years of education between the two groups are related to changes in values. Regarding the impact of sample size, we elucidate that the survey covered the broad urban population of Saint-Louis, including diverse neighborhoods ranging from fishing districts to commercial and working-class areas. To avoid potential bias, randomness was ensured in both surveys using a random walk method to select households. Please refer our responses to the reviewer’s comment for details.

Additional change of manuscript:

Table 2 has been corrected because of a minor error in the calculation of household attribute data. These data were not used for statistical analyses and may not have been affected. Therefore, we have added additional sentences in the abstract and description of the respondent characteristics due to the change in values. In addition, we have changed the method of presenting cumulative values in the expenditure per day per capita comparison to follow the World Bank notation in Table 2.

To clarify the accuracy of the figures, the numbers in Table 3 have been also changed to three decimal places. The author's (YT) affiliation name has been changed to the correct affiliation name. It was the old affiliation name. Table 4 has been corrected minor changed. We sincerely apologize to the editors and reviewers for the inconvenience caused by these changes.

The changed text are below:

In lines 28 to 31 of the Abstract, we have changed the sentences in response to the correction of the calculation of the interaction rate.

[This study suggests that long-term economic insecurity and vulnerable social security have a greater impact on people’s values in Senegal than the threat of pandemic infection, since frequency of people’s social interactions increased compared to the pre-COVID-19 pandemic period.]

In the Discussion section, (line 434 to 437), we altered sentences in response of the correction of calculation in social interaction rate.

[In support of these data, the percentage of respondents in this survey who invited friends, relatives, and others for meals, indicating social interaction, increased after the pandemic compared to that before the pandemic.]

In the Discussion section, we have added new sentences regarding the impact of the global crisis on households. 

---

## [Decision Letter · Decision Letter 1]

1 Aug 2024

PONE-D-23-37707R1Study assesses changes in values of urban Senegalese during global crises

PLOS ONE

Dear Dr. Tobita,

Thank you for submitting your manuscript to PLOS ONE. After careful consideration, we feel that it has merit but does not fully meet PLOS ONE’s publication criteria as it currently stands. Therefore, we invite you to submit a revised version of the manuscript that addresses the points raised during the review process.

We look forward to receiving your revised manuscript.

Kind regards,

Sadia Malik, Ph.D.

Academic Editor

PLOS ONE

Journal Requirements:

Reviewers' comments:

Reviewer's Responses to Questions

**Comments to the Author**

1. If the authors have adequately addressed your comments raised in a previous round of review and you feel that this manuscript is now acceptable for publication, you may indicate that here to bypass the “Comments to the Author” section, enter your conflict of interest statement in the “Confidential to Editor” section, and submit your "Accept" recommendation.

Reviewer #1: All comments have been addressed

Reviewer #2: (No Response)

2. Is the manuscript technically sound, and do the data support the conclusions?

Reviewer #1: Yes

Reviewer #2: No

3. Has the statistical analysis been performed appropriately and rigorously? 

Reviewer #1: Yes

Reviewer #2: No

4. Have the authors made all data underlying the findings in their manuscript fully available?

Reviewer #1: Yes

Reviewer #2: No

5. Is the manuscript presented in an intelligible fashion and written in standard English?

Reviewer #1: Yes

Reviewer #2: No

6. Review Comments to the Author

Reviewer #1: Dear author,

The research on “Study assesses changes in values of urban Senegalese during global crises” presents valuable insights into the dynamic nature of societal values in response to crises such as the COVID-19 pandemic and the Russia-Ukraine War.

The issues related to the practical use of your results by policymakers, health professionals, and others to enhance readiness and reaction to future crises have been resolved. The discussion on the matter of "temporary vs. persistent value changes" has been slightly deepened. The concern of "sampling size and potential biases" has been slightly resolved, the discussion of "other external factors influencing results" has been well discussed. The responses to the reviewers' comments tend to follow logic, and the revisions you made have improved the clarity of your methodology and analytics approach. They also reflect the considered choices you made during the whole process of the research.

Regards,

Reviewer

Reviewer #2: Journal: PLOS ONE

Article title: Study assesses changes in values of urban Senegalese during global crises

Manuscript ID: PONE-D-23-37707R1

General Comments:

This article studies the impact of the COVID-19 pandemic and the Russia-Ukraine war on individual values, focusing on Senegal’s urban population. The authors used the quantitatively assess changes in the values of urban Senegalese during the global crisis with data collected from urban Senegal in August–September 2018 (N=60) and later in June–July 2022 (N=200). The authors reached the conclusions of long-term economic insecurity and vulnerable social security have a greater impact on people’s values in Senegal than the threat of pandemic infection.

Overview:

The paper is very poor written and the empirical work does not appear to be carefully and correctly done. The research question is not good and it does make a sufficient new contribution to the literature to be suitable for the PLOS ONE ONLY after MAJOR revisons.

In fact, the literature on the study assesses changes in values of urban Senegalese during global crises is quite studied much in the literature. The contribution of the paper is the use of the quantitatively assess changes in the values of urban Senegalese during the global crisis with data collected from urban Senegal.

The paper is neutral interesting; and in my view, it needs to be MAJOR improved to reach the standard required for publication in this journal.

Specific Comments:

1. Abstract: somehow theoretical, try to enlarge with the present results from the article

2. Introduction: NOVELTY + results (better explanation);

3. Methodology: why the authors use only these indicators into the model and only this country? Present some theoretical explanations for these indicators

4. Methodology: the main problem of this study is the analysis. The sample of respondents is too small (60 first and 200 after). How the authors choose these people?

5. Methodology: I recommend for authors to read and apply some very serious survey analysis (example: Chapman & Hall - Applied Survey Data Analysis CRC Statistics in the Social and Behavioral Sciences, 2nd Edition)

6. Introduce at least five tables and five figures for the analysis

7. The analysis is very small and sketchy; enlarge this part of the article

8. Analysis: introduce a larger sample for respondents calculated according to the survey analysis

9. Analysis: the survey MUST be on the same respondents in 2019 (before), 2020 (pandemic), 2022 (war) and 2024 after with the same questions. After that, analyze the answers before, during, and after

10. Discussions: at least 2 pages; separate from the conclusions

11. Conclusions: ate least 2 pages, modify and transform into an academic one

General considerations: this article is merely a bachelor thesis, and not a Q2 journal article.

The idea of the article is very mediocre, and the construction of the article is sometimes very subjective. The authors MUST improve the literature, methodology, explanations, discussions, and change the article accordingly. The authors MUST enlarge the data, methodology for the analysis, modify into an academic article.

I ONLY recommend this article be published in PLOS ONE after MAJOR revisions (whole methodology and the discussion).

7. PLOS authors have the option to publish the peer review history of their article (what does this mean?). If published, this will include your full peer review and any attached files.

Reviewer #1: **Yes: **Sara Arzideh

Reviewer #2: No

---

## [Author Response · Author response to Decision Letter 1]

16 Oct 2024

15, October 2024

Dr. Sadia Malik

Academic Editor

PLOS ONE

Dear Dr. Sadia Malik

We wish to re-submit the manuscript titled “Exploring shifts in values among urban Senegalese: The impact of global crises on social and cultural norms.” The manuscript ID is PONE-D-23-37707R2. Old title was “Study assesses changes in values of urban Senegalese during global crises” 

We thank you and the reviewers for your thoughtful suggestions and insights. The manuscript has benefited from these insightful suggestions. I look forward to working with you and the reviewers to move this manuscript closer to publication in the PLOS ONE.

The manuscript has been rechecked and the necessary changes have been made in accordance with the reviewers’ suggestions. The responses to all comments have been prepared and given below.

To facilitate your review of our revisions, the following is a point-by-point response to the questions and comments delivered in your letter, 1 of August, 2024. We have tracked the changes within the manuscript. 

Thank you for your consideration. I look forward to hearing from you.

Sincerely,

Yachiyo Tobita

Ph.D. student, Graduate School of Science and Technology, Degree program in Life and Earth Sciences, University of Tsukuba

1-1-1 Tennoudai, 305-8577 Ibaraki, Japan

Email: yachiyo.tobita@gmail.com

Point-by-point Response to the Reviewer

Comments from Reviewer 2

ABSTRACT

Comment 1: Abstract: somehow theoretical, try to enlarge with the present results from the article.

RESPONSE: Thank you for pointing this out. We agree with this comment. Therefore, we have enlarged with the present result in abstract section (line 23 to 32) as follow;

[Surveys were conducted in Saint-Louis, Senegal, in August-September 2018 and June-July 2022. The timing of these studies coincides with the onset of the COVID-19 pandemic in early 2020 and the outbreak of the Russia-Ukraine war in February 2022. The findings revealed a 19.9% decrease in the average monthly cost of living per capita between 2018 and 2022, attributed to the combined effects of rising food prices and unemployment. Furthermore, the proportion of households spending less than $3.50 per person per day—below the lower-middle-income class poverty line—increased by 11%. Our analysis indicates a decline in values such as benevolence, universalism, hedonism, and self-direction. In contrast, values related to power and achievement significantly increased following the pandemic.]

INTRODUCTION

Comment 2:[ Introduction: NOVELTY + results (better explanation)]

RESPONSE: : Thank you for this insightful comment. We have, accordingly, revised introduction to emphasize the novelty and results in introduction section ( line 63 to 64) 

[Research has utilized quantitative value indicators to examine how external factors, such as global economic crises, influence changes in human values]

In line 81 to 102

[During this period, several global events unfolded, including the emergence of COVID-19, the outbreak of the Russia-Ukraine war, and the resulting global economic turbulence. By comparing values across these two time periods, we aim to shed light on how these historic global events impacted urban Senegalese values. 

There is a notable lack of quantitative studies focusing on the detailed psychological changes among urban Senegalese populations. The novelty of this study lies in its quantitative assessment of changes in values during this period for a specific urban population in Africa, using a standardized value indicator that allows for comparison with other countries and time periods. 

The structure of this paper is as follows: First, we provide an overview of Schwartz’s theory of values, which serves as the study’s theoretical framework. Subsequently, we review studies on individual changes due to COVID-19, using value indicators to highlight the novelty of our research. In the methodology section, we detail the original data collection process. While previous studies indicate that the pandemic led to an increased emphasis on conservative values, the shift in value orientation among urban Senegalese people demonstrated a somewhat different trend. The discussion section analyzes the psychological changes observed in light of our survey findings. Finally, in the conclusion section, we discuss the factors that influenced value changes in Senegalese people, address the limitations of this study, and suggest areas for future research. We believe that by assessing the changes in individual values observed in urban Senegal as a result of the global crises, this study will contribute meaningfully to the growing body of research in this area.]

METHODOLOGY

Comment 3: Methodology: why the authors use only these indicators into the model and only this country? Present some theoretical explanations for these indicators.

RESPONSE: Thank you for this valuable comment. One of the reasons for using this indicator is that we had the opportunity to conduct a comprehensive survey in 2018, prior to the onset of the pandemic and the war, which informed the direction of this study.

We have provided a theoretical explanation regarding the use of this indicator.

(in line 212 to 213)

[In this study, we assessed participants’ values using survey questions from the WVS Wave 6 (2014–2017), based on Schwartz’s ten dimensions value dimensions.]

(in line 224 to 20)

[The value indicators were developed as a standardized method for quantitatively measuring the psychological scales of individuals with varying cultural backgrounds. Numerous international comparative studies, including the WVS, have employed these indicators. The primary advantage of employing these indicators is their standardization, which facilitates future international or intertemporal comparisons. Notably, there is a dearth of value surveys focused on West African countries, and this study aims to contribute to advancing research in this area]

To explained regarding Best Worst Scaling reviews (in line 236-264)

[We used BWS, specifically Case 1, to evaluate respondents’ values. BWS is a survey method designed to measure people’s preferences or the relative importance they assign to multiple items. It has been widely applied in various academic fields in recent years, including health, psychology, and agricultural economics.

In a BWS survey, a subset of all items is presented to respondents, who then choose the most important (best) item and the least important (worst) item from that set. In addition, respondents select one attribute within each question as either the best or worst, making it easier for them to understand and engage with the question. BWS has also been used in surveys based on Schwartz’s theory of basic values. For example, Lee et al. quantitatively measured Schwartz’s ten value items using BWS and highlighted that traditional self-report ratings often suffer from the problem of respondents applying different evaluation criteria. BWS helps reduce this effect by providing a more structured comparison. According to Lee et al., BWS is an effective method for revealing people’s values because it is both more straightforward and relative than traditional methods, such as the Likert scale. Furthermore, Lee et al. conducted a BWS assessment of 19 value items, including new items such as animal welfare, in samples of adults from Australia and the US, demonstrating the utility and reliability of this approach. Daniel et al. used BWS to examine the impact of the pandemic on individual values, employing the counting approach to determine the relative importance of different factors. Similarly, Sneddon et al. used the BWS counting approach to explore changes in individual values related to the natural environment during the pandemic. The results of these studies suggest that the BWS is an effective approach for capturing changes in individual values. To design the survey, we used the support.BWS package in R Core Team version 4.2.2. to create BWS Case 1 questions. The question design followed a Balanced Incomplete Block Design (BIBD), implemented using the crossdes package. Table 2 shows the combinations of value items, each randomly displayed six times using BIBD. As shown in Figure 3, respondents were presented with 11 questions, each consisting of six items, and were asked to select one attribute from the six that was either “very much like me ” or “not at all like me.” ]

Comment 4: Methodology: the main problem of this study is the analysis. The sample of respondents is too small (60 first and 200 after). How the authors choose these people?

RESPONSE: You have raised an important point. While increasing the sample size to a more statistically significant number would have been more appropriate, the sample size in this study is limited. However, to ensure the representativeness of the sample, we have used random sampling techniques, which we believe addresses the concerns regarding participant selection. We have added a detailed explanation of the sampling method and the process for selecting respondents in the Methods section.

In line 289 to 312

[Africa’s demographic transition and rapid urbanization are significantly impacting the lives of its people. Given our emphasis on evolving individual values in urban Africa, we focused on Senegal, which has the highest urbanization rate in West Africa.]

(in line 296 to 305)

[The first survey took place before the pandemic, from late August to early September 2018. After discussions with local collaborators, nine representative districts in Saint-Louis were selected for the study. A random walk method was employed to select six to seven households per district, resulting in a sample of 60 households. 

The second survey was conducted between the post-pandemic period and the Russia-Ukraine War, from early June to late July 2022, using the same random walk method. Of the 33 districts, 10 were selected using probability proportional to size sampling, based on census data provided by the Agence National de la Statistiqué et de la Démographie. A total of 20 households were selected from each district, yielding a total sample of 200 households]

Comment 5:Methodology: I recommend for authors to read and apply some very serious survey analysis (example: Chapman & Hall - Applied Survey Data Analysis CRC Statistics in the Social and Behavioral Sciences, 2nd Edition)

RESPONSE: Thank you for this comments. We agree with this and have incorporated your suggestion throughout the manuscript.

FIGURES AND TABLES

Comment 6: Introduce at least five tables and five figures for the analysis

RESPONSE: We agree with this and have incorporated your suggestion throughout the 

manuscript. To achieve a total of five tables and five figures, three additional figures and one additional table are incorporated.

The following tables and figures are presented:

Introduction section, line104

Figure 1 Change in import broken rice prices in Saint-Louis (2017-2022) 

Figure 2 Theoretical model of the structure of relationships among ten motivational types of value (Schwartz, 2012). 

We add explanation about Figure 2 in line 126 to 127).

[As shown in Figure 2, Schwartz’s theory of basic values outlines ten core human value orientations, which are grouped into four higher-order dimensions ]

Figure 3 Example of choice set for Best Worst Scaling (in line 269)

Figure 4 Survey location in Senegal.(in line 332)

Figure 5 Standardized BW score in 2018 and 2022 surveys.(in line 393)

Table 1 Question for each value attribution based on Schwartz’s theory of basic values.(in line 232)

Table 2 A combination of number of displayed value items in questionnaire. (in line 266)

Table 3 Descriptive statistics of respondents in the 2018 and 2022 surveys.(in line 365)

Table 4 Result of Welch’s t-test.(in line 413)

Table 5 Comparison of values in francophone West African countries.(in line 483)

ANALYSIS

Comment 7: The analysis is very small and sketchy; enlarge this part of the article

RESPONSE: Thank you for pointing this out. We agree with this comment. Therefore, we have enlarged and explained in line 372 to 379.

[Based on the aggregated standardized BW scores from the 2018 and 2022 surveys, we compared the responses for each value item. Figure 5 displays the mean standardized scores for each value item. A positive score indicates that respondents consider the value important, while a negative score suggests the opposite. This comparison highlights respondents’ value tendencies over time. In both surveys, tradition—representing family and religious values— consistently ranked as the most important. Respondents also emphasized benevolence-caring, symbolizing kindness toward others, and conformity, reflecting social discipline. These values correspond to the conservation and self-transcendence dimensions of values.]

We add in line 400 to 406.

[Table 4 presents the results of statistical tests conducted to evaluate the differences in the mean values for each item between 2018 and 2022. Given the unequal sample sizes in the two surveys, a standard t-test assuming equal variances was not applicable. Instead, the Welch test was employed to account for these differences. A two-tailed test was conducted with the null hypothesis that there were no significant changes in the values of respondents between 2018 and 2022. If a significant difference in the mean values was found, the null hypothesis was rejected.]

In line 444 to 457, we enlarged this part and add subsection summary.

[Using the Welch test, we examined the changes in values and identified those with significant differences. The overall trend in values remained consistent across 2018 and 2022. Tradition (from the conservation group), along with benevolence-caring and benevolence-dependability (from the self-transcendence group), were identified as the most important values in both years. However, achievement and power (self-enhancement values), which were not deemed important in 2018, became more significant in 2022. Conversely, the importance of hedonism and self-direction (openness to change values) declined significantly. 

The examination of changes in standardized BW scores across the two time points revealed several values with significant differences. Power and achievement, which reflect financial success and social status, showed a significant increase in scores. In contrast, self-direction and hedonism, values associated with creativity and personal enjoyment, experienced a decline. Additionally, scores for benevolence-caring, benevolence-dependability, and universalism—values tied to environmental and societal contributions—also decreased significantly.]

Comment 8: Analysis: introduce a larger sample for respondents calculated according to the survey analysis.

RESPONSE: Thank you for your comments. You have raised an important point here. However, the dataset analyzed in the manuscript was obtained from the original survey conducted in 2018 and 2022. Consequently, it is not feasible to conduct further surveys using data from previous years. We believe that using these samples is a more appropriate approach given the unfeasibility of expanding the sample size of the dataset.

Comment 9:Analysis: the survey MUST be on the same respondents in 2019 (before), 2020 (pandemic), 2022 (war) and 2024 after with the same questions. After that, analyze the answers before, during, and after

RESPONSE: Thank you for this suggestion. It would have been interesting to explore this aspect. However, due to the numerous restrictions that were in place during the period of the COIVD-19 pandemic and in the subsequent period, it would have been challenging to achieve this within the scope of our study. Furthermore, we would like to clarify that this is not an online survey. The two surveys were designed via face-to-face interviews conducted over five consecutive days to collect data. Due to the various restrictions imposed by the circumstances of the ongoing pandemic and the need to ensure the safety of our surveyors, we were unable to proceed with the survey, particularly during the 2020 and 2021 periods. 

DISCUSSION

Comment 10: Discussions: at least 2 pages; separate from the conclusions

RESPONSE: Thank you for the sug

---

## [Decision Letter · Decision Letter 2]

9 Dec 2024

Exploring shifts in values among urban Senegalese: The impact of global crises on social and cultural norms

PONE-D-23-37707R2

Dear Dr. Tobita,

We’re pleased to inform you that your manuscript has been judged scientifically suitable for publication and will be formally accepted for publication once it meets all outstanding technical requirements.

Kind regards,

Carolyn Chisadza

Academic Editor

PLOS ONE

Additional Editor Comments (optional):

Reviewers' comments:

Reviewer's Responses to Questions

**Comments to the Author**

1. If the authors have adequately addressed your comments raised in a previous round of review and you feel that this manuscript is now acceptable for publication, you may indicate that here to bypass the “Comments to the Author” section, enter your conflict of interest statement in the “Confidential to Editor” section, and submit your "Accept" recommendation.

Reviewer #2: All comments have been addressed

2. Is the manuscript technically sound, and do the data support the conclusions?

Reviewer #2: Yes

3. Has the statistical analysis been performed appropriately and rigorously? 

Reviewer #2: Yes

4. Have the authors made all data underlying the findings in their manuscript fully available?

Reviewer #2: Yes

5. Is the manuscript presented in an intelligible fashion and written in standard English?

Reviewer #2: Yes

6. Review Comments to the Author

Reviewer #2: Journal: PLOS ONE

Article title: Exploring shifts in values among urban Senegalese: The impact of global crises on social and cultural norms

Manuscript ID: PONE-D-23-37707R2

Dear Author (s);

Dear Editor,

The manuscript has been revised for better interpretations according to the suggestions of the reviewer(s) by including the information required.

The author(s) change the interpretations, results, methodology, and conclusions accordingly, and therefore, the paper is much improved now. The author(s) reduces considerably the article, references, and diversifies the articles cited.

I recommend that this article be published in PLOS ONE.

Congratulations!

7. PLOS authors have the option to publish the peer review history of their article (what does this mean?). If published, this will include your full peer review and any attached files.

Reviewer #2: No

---

## [Editor Report · Acceptance letter]

12 Jan 2025

PONE-D-23-37707R2 

PLOS ONE

Dear Dr. Tobita, 

I'm pleased to inform you that your manuscript has been deemed suitable for publication in PLOS ONE. Congratulations! Your manuscript is now being handed over to our production team.

Kind regards, 

on behalf of

Prof Carolyn Chisadza 

Academic Editor

PLOS ONE
